# Neural ePDOs: Spatially Adaptive Equivariant Partial Differential Operator Based Networks

**Lingshen He[1], Yuxuan Chen[2],[\*] Zhengyang Shen[3], Yibo Yang[2], Zhouchen Lin[1,4,5][†]**

[1] National Key Lab of General AI, School of Intelligence Science and Technology, Peking University
[2] JD Explore Academy, Beijing, China
[3] Department of Computer Vision Technology (VIS), Baidu Inc.
[4] Institute for Artificial Intelligence, Peking University
[5] Peng Cheng Laboratory

## Abstract

Endowing deep learning models with symmetry priors can lead to a considerable performance improvement. As an interesting bridge between physics and deep learning, the equivariant partial differential operators (PDOs) have drawn much researchers' attention recently. However, to ensure the PDOs translation equivariance, previous works have to require coefficient matrices to be constant and spatially shared for their linearity, which could lead to the sub-optimal feature learning at each position. In this work, we propose a novel nonlinear PDOs scheme that is both spatially adaptive and translation equivariant. The coefficient matrices are obtained by local features through a generator rather than spatially shared. Besides, we establish a new theory on incorporating more equivariance like rotations for such PDOs. Based on our theoretical results, we efficiently implement the generator with an equivariant multilayer perceptron (EMLP). As such equivariant PDOs are generated by neural networks, we call them Neural ePDOs. In experiments, we show that our method can significantly improve previous works with smaller model size in various datasets. Especially, we achieve the state-of-the-art performance on the MNIST-rot dataset with only tenth of parameters of the previous best model.

## 1 Introduction

In recent years, convolutional neural networks (CNNs) have achieved superior performance on various vision tasks (Szegedy et al., 2015; He et al., 2016; Chen et al., 2017). It is acknowledged that the success of CNNs is attributed to their ability to exploit the intrinsic translation-invariance symmetry of data to help downstream vision tasks. To incorporate other symmetries like rotation-invariance, various CNNs-based equivariant networks have been studied and carried out to enhance the performance of vision tasks (Cohen & Welling, 2016a;b; Weiler & Cesa, 2019). In another branch, some works (Osher & Rudin, 1990; Perona & Malik, 1990) adopted partial differential operators (PDOs) to process images in the early period. Recently, PDOs with learnable coefficients are adopted by Shen et al. (2020) to design equivariant networks which achieve competitive performance compared to previous equivariant networks. Jenner & Weiler (2021) further generalized this work to a unified framework on the equivariant linear PDOs on Euclidean spaces of various representation types.

Actually, the coefficient matrices of the current PDOs works are spatially shared, e.g. the same PDOs are applied to process features at each position (see Figure.1(a)). However, such a coefficient sharing scheme of the PDOs is not the optimal pattern to extract features from input images (Wu et al., 2018; Su et al., 2019; Zhou et al., 2021; He et al., 2021a). To be specific, the contents of the input images vary according to positions, e.g. some pixels cover the background while some express texture, which would make coefficient-sharing PDOs inefficient to extract features at each position.

---

[\*]Equal first authorship
[†]Corresponding author

In fact, Jenner & Weiler (2021) have proved that the linear PDOs layer is translation equivariant if and only if its coefficient matrices are spatially shared, so it seems impossible to ensure both the spatial adaptivity and translation equivariance for PDOs.

In this work, to deal with the above issue, we think outside the box of the linear limitation and propose brand new nonlinear PDOs that are both spatially adaptive and translation equivariant. Compared with spatially shared PDOs, we construct a coefficient generator that inputs local features and outputs the coefficient matrices. Since different positions produce different coefficient matrices, the PDOs are essentially position-specific and can extract individual features according to the local content (see Figure 1(b)). In addition, the coefficient matrices generated by local features guarantee the translation equivariance for such PDOs naturally. However, such a nonlinear PDOs scheme is not intrinsically equivariant to rotations or reflections. To incorporate equivariance of these transformations, we establish a theory on the equivariant formulation of this nonlinear PDOs scheme under any given symmetry group. Specifically, the theory reveals that *this type of PDOs is equivariant if and only if the coefficient generators are exactly equivariant maps of particular transformations*. In practice, we choose a two-layer EMLP (Finzi et al., 2021) as the coefficient generator to satisfy the equivariance condition and provide an efficient implementation scheme. We name our model Neural ePDOs and evaluate its performance on MNIST-rot and ImageNet datasets. Extensive experiments show that our model can significantly improve accuracy with fewer parameters. Especially, we achieve the state-of-the-art results on MNIST-rot dataset with only a tenth of the parameters compared to previous best models.

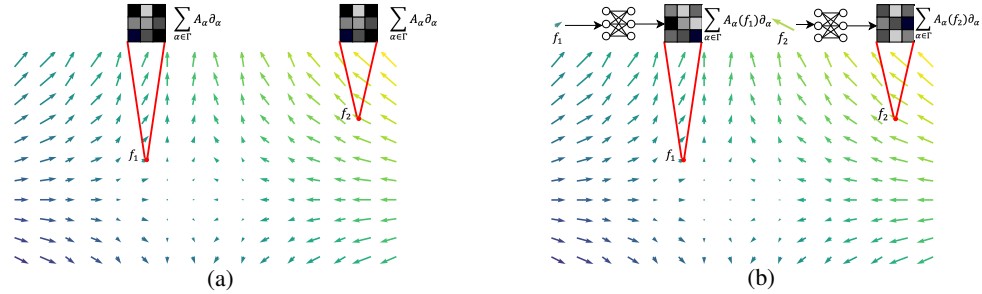

Figure 1: Illustration of two different designs for PDOs. Here, we use the 2-dimensional vector field to represent the feature map. (a)For linear PDOs, the coefficient matrices are shared to process features across different positions. (b) For nonlinear PDOs we propose in this paper, the coefficient matrices are generated by the local features through neural networks.

We summarize the main contributions as follows:

- To our knowledge, we are the first one to propose the nonlinear form of PDOs that are both spatially adaptive and translation equivariant. The coefficient matrices of the novel PDOs are adaptive to local features, which could alleviate the sub-optimal feature learning problem at each position.
- We develop a theory for such nonlinear PDOs that precisely characterize when it is equivariant under any given symmetry group. The theory reveals that the nonlinear PDOs are equivariant if and only if the coefficient generators are exactly equivariant maps of particular transformations.
- We provide an efficient implementation which adopts a two-layer EMLP as the coefficient generator and could largely save parameters and computations.
- Extensive experiments show that our method can significantly improve the results on MNIST-rot and ImageNet datasets with significantly fewer parameters. Especially, we achieve state-of-the-art results on the MNIST-rot dataset.

## 2 RELATED WORKS

So far, there are two mainstream approaches to constructing group equivariant networks. One is first developed by Cohen & Welling (2016a) which views the feature maps as maps defined on a

group, and they proposed group convolution operation to process these feature maps equivariantly for image recognition. The method is further applied to designing equivariant networks for 3D space (Worrall & Brostow, 2018), sphere (Cohen et al., 2018), video tracking (Gupta et al., 2021) and lie groups (Finzi et al., 2020a; Bekkers, 2019), etc. This approach is further developed to design attentive convolution layer (Romero et al., 2020) and self-attention layer (Romero & Cordonnier, 2020; Hutchinson et al., 2021; He et al., 2021b).

The other one follows the approach of steerable CNNs (Cohen & Welling, 2016b; Weiler & Cesa, 2019; Weiler et al., 2018; Jenner & Weiler, 2021), which is a generalization of the first approach. Analogous to physics, the feature map here is viewed as a field, which is transformed according to a specified group representation under the act of transformation. In comparison, the feature map in the first approach is simply the field with regular representation. Works (Cohen & Welling, 2016b; Weiler et al., 2018; Weiler & Cesa, 2019) are devoted to finding out all the equivariant convolution operations as a map between any two fields. Later works further generalize the approach to design equivariant transformer (Fuchs et al., 2020) and graph network (Brandstetter et al., 2021). Our work follows these approaches.

Recently, some works focus on utilizing PDOs to design equivariant neural networks, as they can build an interesting bridge between physics and deep learning (Jenner & Weiler, 2021). In addition, PDOs are very suitable for processing continuous data (Finzi et al., 2020b) and non-Euclidean structure data. The work most closely related to ours is Jenner & Weiler (2021) which derives steerable PDOs as a linear map between any two fields in the language of group representation theory. It is an extension of PDO-eConv (Shen et al., 2020) which employs rotated PDOs to design linear equivariant layers similar to the approach of Cohen & Welling (2016a). In our work, to alleviate the spatial-agnostic problem in linear PDO-based equivariant layers, we propose a nonlinear PDO scheme and develop an equivariant theory that generalizes the Jenner & Weiler (2021). A more detailed comparison between our work and steerable PDOs can be found in supplementary material.

## 3 PRELIMINARY

### 3.1 EQUIVARIANCE

Equivariance measures how the output of a network layer transforms in a predictable way with respect to the transformation of the input. In mathematics, a map $\Psi$ is group equivariant if it satisfies:

$$\forall h \in H, \quad \Psi\left[\pi(h)[\mathbf{f}]\right] = \pi'(h)[\Psi[\mathbf{f}]], \tag{1}$$

where $H$ is a transformation group, $\pi(h)$ and $\pi'(h)$ are group actions, and $\mathbf{f}$ is the input. In CNNs, $\mathbf{f}$ is the feature map which can be seen as a vector-valued function $\mathbf{f} : \mathbb{R}^2 \to \mathbb{R}^n$, where $\mathbb{R}^n$ is the $n$-dimensional vector space. If we choose $H$ to be the translation group, it is easy to prove that the convolution layer satisfies this requirement. In the following, we mainly consider feature maps defined on $\mathbb{R}^2$, and the conclusions can be readily extended to feature maps defined on any dimension.

Following the standard practice of equivariant deep learning, the feature map $\mathbf{f}$ is modeled as a vector field composed of fiber $\mathbf{f}(x)$ located at every point $x \in \mathbb{R}^2$. For transformation group $H$, we mainly consider affine group of the form $H = (\mathbb{R}^2, +) \rtimes G$, for some $G \leq \text{GL}(2, \mathbb{R})$. Here, $H$ is constructed by the semi-direct product between translation group $(\mathbb{R}^2, +)$ and a linear invertible transformation group $G$ performed on $\mathbb{R}^2$, e.g. rotations and mirrorings. The group action $\pi(h)$ acts on field $\mathbf{f}$ as:

$$\forall x \in \mathbb{R}^2, \quad \pi(h)\mathbf{f}(x) = \boldsymbol{\rho}(g)\mathbf{f}(g^{-1}(x - t)), \tag{2}$$

where $t \in \mathbb{R}^2$ is a translation, $g \in G$ is a linear transformation, $h := (t, g) \in H$, and $\boldsymbol{\rho}(g)$ is a group representation of $g$. Formally, a group representation $\boldsymbol{\rho}$ of the group $G$ is a group homomorphism: $G \to \mathbb{R}^{n \times n}$, i.e., $\forall g_1, g_2 \in G, \boldsymbol{\rho}(g_1 g_2) = \boldsymbol{\rho}(g_1)\boldsymbol{\rho}(g_2)$. It describes how each fiber transforms under the group action. When $\boldsymbol{\rho}$ is given, its corresponding feature map is called a $\boldsymbol{\rho}$-field. See supplementary material for more introduction to group representations.

### 3.2 PARTIAL DIFFERENTIAL OPERATORS

Partial differential operators (PDOs) are commonly used in physics areas, such as gradient, curl or Laplacian. They can be seen as a kind of maps between smooth functions. Given a smooth $c_{in}$-

dimensional feature map $\mathbf{f} = (f_1, ..., f_{c_{in}})^T$, the PDO $\partial_x$ acts on $f$ as $\partial_x \mathbf{f} := (\partial_x f_1, ..., \partial_x f_{c_{in}})^T$. In general, the PDOs can be formalized as the linear combination of various orders of elementary PDO which is denoted as $\partial^\alpha := \partial_{x_1}^{\alpha_1} \partial_{x_2}^{\alpha_2}$, $\alpha = (\alpha_1, \alpha_2) \in \mathbb{N}_0^2$. Here, we adopt the multi-index notation on elementary PDOs as Jenner & Weiler (2021). For simplicity, we utilize $\Gamma_N = \{(i,j) | i, j \in \mathbb{N}_0, 0 \le i + j \le N\}$ to index elementary PDOs with their order less than $N$ as we have to set a truncation order to implement PDOs in the computer. For example, the PDOs from $C^\infty(\mathbb{R}^2, \mathbb{R}^{c_{in}})$ to $C^\infty(\mathbb{R}^2, \mathbb{R}^{c_{out}})$ with truncation order $N = 3$ can be formalized as

$$
\begin{aligned}
\hat{D}^{(3)}\mathbf{f} := & \mathbf{W}_{(0,0)}\mathbf{f} + \mathbf{W}_{(1,0)}\partial^{(1,0)}\mathbf{f} + \mathbf{W}_{(0,1)}\partial^{(0,1)}\mathbf{f} + \mathbf{W}_{(2,0)}\partial^{(2,0)}\mathbf{f} + \mathbf{W}_{(1,1)}\partial^{(1,1)}\mathbf{f} \\
& + \mathbf{W}_{(0,2)}\partial^{(0,2)}\mathbf{f} + \mathbf{W}_{(3,0)}\partial^{(3,0)}\mathbf{f} + \mathbf{W}_{(2,1)}\partial^{(2,1)}\mathbf{f} + \mathbf{W}_{(1,2)}\partial^{(1,2)}\mathbf{f} + \mathbf{W}_{(0,3)}\partial^{(0,3)}\mathbf{f},
\end{aligned}
\tag{3}
$$

where $\mathbf{W}_{(i,j)} : \mathbb{R}^2 \to \mathbb{R}^{c_{out} \times c_{in}}$ is the coefficient matrix corresponding to $\partial^{(i,j)}$.

To study the equivariance of PDOs, we give a description of the transformation property of elementary PDOs. Here, we assume the input $\mathbf{f}$ to be scalar field and take $\partial^{(2,0)}$ as an example. When input of the PDO go through a affine transformation $(g, t)$, resulting in $\tilde{\mathbf{f}}(x) := \mathbf{f}(g^{-1}(x - t))$. According to the chain rule, we get:

$$
[\partial^{(2,0)}\tilde{\mathbf{f}}](x) = \left(g_{11}^{-1}\right)^2 [\partial^{(2,0)}\mathbf{f}](\tilde{x}) + 2g_{11}^{-1}g_{21}^{-1}[\partial^{(1,1)}\mathbf{f}](\tilde{x}) + \left(g_{21}^{-1}\right)^2 [\partial^{(0,2)}\mathbf{f}](\tilde{x}),
\tag{4}
$$

where $g_{11}^{-1}$ and $g_{21}^{-1}$ are matrix elements of $g^{-1}$ and $\tilde{x} := g^{-1}(x - t)$. In general, for each elementary PDO:

$$
\forall \alpha \in \Gamma_N, x \in \mathbb{R}^2, g \in G, \quad \partial^\alpha \tilde{\mathbf{f}}(x) = \sum_{\beta \in \Gamma_N} \hat{\rho}_{\alpha,\beta}(g)\partial^\beta \mathbf{f}(\tilde{x}),
\tag{5}
$$

where $\hat{\rho}_{\alpha,\beta}(g)$ denotes the transformation coefficient in front of elementary PDO $\partial^\beta$ on the right side of the above equation of a given $\alpha$. All these transformation coefficients of a given group element $g$ constitute a matrix $\hat{\boldsymbol{\rho}}(g)$. We have the following result:

**Lemma 1** $\hat{\boldsymbol{\rho}}(g)$ *defined in Eq.(5) is a group representation of g on* $\mathbb{R}^{|\Gamma_N|}$.

Proof of the lemma can be found in supplementary material. We also give a procedure in the supplementary material to automatically compute $\hat{\boldsymbol{\rho}}(g)$ for any given $N$.

## 4 THE NEURAL EPDOS FRAMEWORK

In this section, we first propose the new scheme of PDOs, in which the coefficient matrices are generated by features. Then, we propose the general theory that gives a necessary and sufficient condition to ensure equivariance for the operator for any given symmetry. As coefficient generators will induce heavy parameters and computation costs, we propose to require coefficient matrices to be diagonal and characterize the equivariant space for it.

### 4.1 A NONLINEAR PDOS SCHEME

As introduced in Section 3.2, PDOs as maps from $C^\infty(\mathbb{R}^2, \mathbb{R}^{c_{in}})$ to $C^\infty(\mathbb{R}^2, \mathbb{R}^{c_{out}})$ are formulated as :

$$
\forall x \in \mathbb{R}^2, \quad \Psi[\mathbf{f}](x) = \sum_{\alpha \in \Gamma_N} \mathbf{W}_\alpha(x)\partial^\alpha[\mathbf{f}](x),
\tag{6}
$$

where $\mathbf{f} \in C^\infty(\mathbb{R}^2, \mathbb{R}^{c_{in}})$, $\mathbf{W}_\alpha : \mathbb{R}^2 \to \mathbb{R}^{c_{out} \times c_{in}}$. Jenner & Weiler has proved that the necessary and sufficient condition for Eq.(6) to be translation equivariant is to require the coefficients $\mathbf{W}_\alpha$ to be spatially shared, that is, $\forall x, x' \in \mathbb{R}^2, \alpha \in \Gamma_N, \mathbf{W}_\alpha(x) = \mathbf{W}_\alpha(x')$. However, it is not efficient for the spatially shared PDOs to learn diverse patterns in the feature map, which may lead to the redundancy of learnable parameters.

To alleviate this problem, we propose a nonlinear PDOs scheme that adjusts PDOs according to features at different positions. Furthermore, our newly proposed module still keeps translation equivariance as the spatially shared PDOs. Specifically, we adopt $\mathbf{W}_\alpha$ as the coefficient generators to generate coefficient matrices from local input features, which can be formulated as:

$$
\forall x \in \mathbb{R}^2, \quad \Psi[\mathbf{f}](x) = \sum_{\alpha \in \Gamma_N} \mathbf{W}_\alpha(\mathbf{f}(x))\partial_\alpha[\mathbf{f}](x),
\tag{7}
$$

where $\mathbf{W}_\alpha : \mathbb{R}^{c_{in}} \to \mathbb{R}^{c_{out} \times c_{in}}, \alpha \in \Gamma_N$ are coefficient generators with local features as input. It is easy to check the translation equivariance of Eq.(7). We could adopt MLP as the structure of the coefficient generators as they are the universal approximator of any continuous function. Then, the neural network and local input features decide the specific PDOs applied at each position.

## 4.2 EQUIVARIANCE THEORY

Although the nonlinear PDOs formulated in Eq.(7) are equivariant to translation, they are not intrinsically equivariant to common transformations such as rotation or reflection. We now derive a complete characterization of their equivariant space of such symmetry.

The equivariant requirement on the operators (7), in the sense defined by Eq.(1), can be reduced to the requirement on the coefficient generators $\mathbf{W}_\alpha$. Supposing the input and output of the operator are any $\boldsymbol{\rho}$-field and $\boldsymbol{\rho}'$-field, respectively, we have,

**Proposition 1** *The nonlinear PDOs in Eq.(7) are equivariant to affine transformation H if and only if the coefficient generators satisfy the following constraint:*

$$\forall \alpha \in \Gamma_N, \forall g \in G, \forall \mathbf{y} \in \mathbb{R}^{c_{in}}, \quad \sum_{\beta \in \Gamma_N} \hat{\rho}_{\beta,\alpha}(g)\mathbf{W}_\beta(\boldsymbol{\rho}(g)\mathbf{y})\boldsymbol{\rho}(g) = \boldsymbol{\rho}'(g)\mathbf{W}_\alpha(\mathbf{y}). \tag{8}$$

The proof of this proposition can be found in supplementary material. The proof makes use of the fact that elementary PDOs are independent of each other. It is remarkable that the above equation constraints for coefficient generators are imposed for each $\alpha \in \Gamma_N$. To uncover the intrinsic structure of $\mathbf{W}_\alpha$, we further concatenate all the coefficient generators side by side as a whole $\mathbf{W}$ such that $\forall \mathbf{y} \in \mathbb{R}^{c_{in}}, \mathbf{W}(\mathbf{y}) = [\mathbf{W}_{(0,0)}(\mathbf{y}), ..., \mathbf{W}_{(0,N)}(\mathbf{y})]$. Then, the constraint Eq.(8) reduces to the following form:

**Proposition 2** *Eq.(8) is equivalent to the following form:*

$$\forall g \in G, \forall \mathbf{y} \in \mathbb{R}^{c_{in}}, \quad \mathbf{W}(\boldsymbol{\rho}(g)\mathbf{y})(\hat{\boldsymbol{\rho}}(g) \otimes \boldsymbol{\rho}(g)) = \boldsymbol{\rho}'(g)\mathbf{W}(\mathbf{y}). \tag{9}$$

*Here, $\otimes$ is the tensor product of two group representations.*

We show the proof at supplementary material. Applying the vec-operator[1] on Eq.(9), it can be rewritten as:

$$\forall g \in G, \forall \mathbf{y} \in \mathbb{R}^{c_{in}}, \quad \mathrm{vec}[\mathbf{W}(\boldsymbol{\rho}(g)\mathbf{y})] = (\boldsymbol{\rho}'(g) \otimes \hat{\boldsymbol{\rho}}(g^{-1})^\top \otimes \boldsymbol{\rho}(g^{-1})^\top)\mathrm{vec}[\mathbf{W}(\mathbf{y})], \tag{10}$$

where $\mathrm{vec}[\cdot]$ operator flattens the matrix into a vector by concatenating the rows of a matrix one by one. Eq.(10) reveals that $\mathrm{vec}[\mathbf{W}]$ is an equivariant function with input and output vector transform according to $\boldsymbol{\rho}(g)$ and $\boldsymbol{\rho}'(g) \otimes \hat{\boldsymbol{\rho}}(g^{-1})^\top \otimes \boldsymbol{\rho}(g^{-1})^\top$, respectively. It is easy to check that $\boldsymbol{\rho}'(g) \otimes \hat{\boldsymbol{\rho}}(g^{-1})^\top \otimes \boldsymbol{\rho}(g^{-1})^\top$ is also a representation of $G$. As we adopt coefficient generators as MLP, $\mathrm{vec}[\mathbf{W}]$ is an EMLP and can be efficiently constructed by Finzi et al. (2021).

As coefficient matrices in such operators are generated via MLP, it brings much extra computational burden compared to steerable PDOs. To alleviate the problem, we propose a novel structure for coefficient generators and give its equivariant characterization in the following.

## 4.3 EFFICIENT COEFFICIENT GENERATORS

In practice, regular representation and quotient representation (see supplementary material) are mostly adopted for equivariant networks (Weiler & Cesa, 2019) due to their superior performance. Therefore, we attempt to propose an efficient coefficient generator for these representation types in this subsection. In Eq.(7), directly generating the whole coefficient matrix $\mathbf{W}$ would make it suffer heavy parameter and computation costs. To alleviate this issue, we assume that the coefficient matrices in Eq.(7) are diagonal matrices. Then we can formulate the operator (7) as:

$$\forall x \in \mathbb{R}^2, \quad \Psi[\mathbf{f}](x) = \sum_{\alpha \in \Gamma_N} \mathbf{w}_\alpha(\mathbf{f}(x)) \circ \partial_\alpha[\mathbf{f}](x), \tag{11}$$

---

[1]For matrices $\mathbf{A}, \mathbf{X}, \mathbf{B}$, we have $\mathrm{vec}(\mathbf{A}\mathbf{X}\mathbf{B}) = (\mathbf{A} \otimes \mathbf{B}^\top)\mathrm{vec}(\mathbf{X})$.

where $\mathbf{w}_\alpha : \mathbb{R}^{c_{in}} \to \mathbb{R}^{c_{in}}$ and $\circ$ is used to denote element-wise product between two vectors. Here, we have assumed output and input to be of the same field type and, if necessary, we can follow it with a linear projection to transform it into another field. Such design greatly reduces the computational burden for generating coefficients and also works well in the experiment in Section 7.

According to Proposition 1, the above operator (11) is equivariant if and only if:

$$\forall \alpha \in \Gamma_N, \forall g \in G, \forall \mathbf{y} \in \mathbb{R}^{c_{in}}, \quad \sum_{\beta \in \Gamma_N} \hat{\rho}_{\beta,\alpha}(g)\text{diag}[\mathbf{w}_\beta(\boldsymbol{\rho}(g)\mathbf{y})]\boldsymbol{\rho}(g) = \boldsymbol{\rho}(g)\text{diag}[\mathbf{w}_\alpha(\mathbf{y})]. \quad (12)$$

We use $\text{diag}[\cdot]$ to denote converting an $n$-dimensional vector to an $n$-dimensional diagonal matrix with the vector as diagonal. Because of the diagonal operation, directly using results in Proposition 2 cannot uncover the structure of $\mathbf{w}_\alpha$ satisfying the above constraint. By utilizing the special structure of regular representation and quotient representation, we have the following result.

**Proposition 3** *Suppose the input and output of operator (11) are both $\boldsymbol{\rho}(g)$-field. If the $\boldsymbol{\rho}$ is a regular or quotient representation of $G$, the constraint in Eq.(12) is equivalent to:*

$$\forall g \in G, \forall \mathbf{y} \in \mathbb{R}^{c_{in}}, \quad \bar{\mathbf{w}}(\boldsymbol{\rho}(g)\mathbf{y}) = (\boldsymbol{\rho}(g) \otimes \hat{\boldsymbol{\rho}}(g^{-1})^\top)\bar{\mathbf{w}}(\mathbf{y}), \quad (13)$$

where $\forall \mathbf{y} \in \mathbb{R}^{c_{in}}, \bar{\mathbf{w}}(\mathbf{y}) = \text{vec}([\mathbf{w}_{(0,0)}(\mathbf{y}), ..., \mathbf{w}_{(0,N)}(\mathbf{y})])$ is a large vector concatenated from all generated vectors (see supplementary material for proof). Similar to the above general case, $\bar{\mathbf{w}}$ is an equivariant function with input and output vector transform according to $\boldsymbol{\rho}(g)$ and $(\boldsymbol{\rho}(g) \otimes \hat{\boldsymbol{\rho}}(g^{-1})^\top)$, respectively. So far, we have fully characterized the structure of the coefficient generators which ensures operator (11) is equivariant. As the coefficient generator is based on an equivariant neural network, we name our model Neural ePDOs. In the next section, we will show a detailed implementation that is both parameters efficient and computationally efficient.

# 5 IMPLEMENTATION OF NEURAL EPDOS

## 5.1 DESIGN OF COEFFICIENT GENERATOR

As is shown in Proposition 3, coefficient generator $\bar{\mathbf{w}}$ can be viewed as an equivariant vector-valued function. In practice, we implement it as an EMLP which can be constructed as Finzi et al. (2021). In this paper, we adopt a two-layer EMLP as $\bar{\mathbf{w}}$. To reduce parameter and computation costs, we choose a bottleneck design here, a relatively small size of the hidden layer. Specifically, $\bar{\mathbf{w}}(x) = \mathbf{W}^2\text{ReLu}(\mathbf{W}^1 x)$, where $\mathbf{W}^1 \in \mathbb{R}^{c_{mid} \times c_{in}}$, $\mathbf{W}^2 \in \mathbb{R}^{|\Gamma_N|c_{in} \times c_{mid}}$, $c_{mid} = \frac{c_{in}}{r}$, where $r$ is the reduction ratio, and $\text{ReLu}(\cdot)$ is a element-wise ReLu activation function (Nair & Hinton, 2010). Here, we assume $\boldsymbol{\rho} = p\boldsymbol{\rho}_0$, in other words, representation $\boldsymbol{\rho}$ can be decomposed into multiple identical representations $\boldsymbol{\rho}_0$, which is very common in practice. Then the output representation of $\bar{\mathbf{w}}(x) = \mathbf{W}^2\text{ReLu}(\mathbf{W}^1 x)$ can be decomposed in a similar way, *i.e.*, $(\boldsymbol{\rho} \otimes (\hat{\boldsymbol{\rho}}^{-1})^\top) = p(\boldsymbol{\rho}_0 \otimes (\hat{\boldsymbol{\rho}}^{-1})^\top)$. We reduce both computations and parameters by requiring the outputs of the $p$ partitions to be the same.

## 5.2 DISCRETIZATION OF PDOS

Our theory for Neural ePDOs is developed in the continuous space. In practice, in order to process digital images which are defined on two-dimensional grids, we need to discretize our PDOs. In our paper, we mainly consider the finite difference (FD) method and the Gaussian derivatives method (GA) (Jenner & Weiler, 2021). In principle, we only need to consider the discretization of the elementary PDO because the discretization of the whole PDOs can be obtained by a linear combination of them.

**FD**: Finite difference method is widely used in numerical analysis to approximate PDO by a linear combination of function values on finite grids. For example, $\partial_x f = (f(x+1) - f(x-1))/2$. On the regular grids, a PDO can be approximated by a convolution operation (Shen et al., 2020), *i.e.*,

$$\partial^\alpha[f] \approx u_\alpha * \mathbf{F}, \quad (14)$$

where $u_i$ is a convolution filter. Corresponding filters for elementary PDOs in $P$ are provided in the supplementary material.

**GA**: PDO can also be estimated by taking derivatives of Gaussian function (Jenner & Weiler, 2021), *i.e.,* given grid points $x_n \in \mathbb{R}^2$, $\forall \alpha \in \Gamma_N$,

$$\partial^\alpha[f] \approx \partial^\alpha[G(x_n; \sigma)]f(x_n), \tag{15}$$

where $G(x; \sigma)$ is a Gaussian kernel with standard deviation $\sigma$ around 0.

## 6   COMPLEXITY ANALYSIS

In this section, we give a complexity analysis for both steerable PDOs and Neural ePDOs. For simplicity, we assume the feature field to be a regular field (Results for other feature fields are similar). We especially consider the complexity for Neural ePDOs with full coefficient matrices (denoted as full) and diagonal coefficient matrices (denoted as diag), respectively. Here, we assume the EMLP of Neural ePDOs (full) to have the same design as Neural ePDOs (diag). Suppose the representations type of the input feature map and output feature map to be $c\rho_{reg}$. The width and height of the feature map are $h$ and $w$, respectively. $n$ is the number of elements in the group $G$ and $k$ is the discretization kernel size. Both parameters and flops complexity are listed in Table (1).

We first make a comparison of parameters. For both the Neural ePDOs (full) and Neural ePDOs (diag), the first term is the number of parameters in the first layer of EMLP (coefficient generator) and the second term is for the second layer. It is obvious that Neural ePDOs (full) have significantly more parameters than Neural ePDOs (diag) ($O(c^3)$ vs $O(c^2)$). For comparison of parameters between steerable PDOs and Neural ePDOs (diag), both the first term and second term in Neural ePDOs (diag) are much less than the parameters of steerable PDOs (In our paper, we set $N = 4$), hence Neural ePDOs (diag) is much parameter efficient than steerable PDOs.

For flops, the first two terms in the Neural ePDOs (both full and diag) are used for coefficient generation and the last term is used for the action of PDOs. As the last term in Neural ePDOs (full) is equal to steerable PDOs, the latter one surely requires less computational burden. It is also easy to check all three terms in the Neural ePDOs (diag) are much less than flops of steerable PDOs, hence Neural ePDOs (diag) is much more computationally efficient than steerable PDOs.

From the comparison above, both the diagonal restriction and EMLP design (bottleneck structure (r) and partition (p) operation) help to make our Neural ePDOs more efficient than steerable PDOs.

Table 1: Complexity Analysis of each PDO layer.

| Method | Parameters | Flops |
|---|---|---|
| Steerable PDO | $c^2(N+1)(N+2)n/2$ | $c^2n^2k^2hw$ |
| Neural ePDOs (full) | $c^2n/r + c^3(N+1)(N+2)n/2rp$ | $(c^2n^2/r + c^3n^3k^2/rp)hw + c^2n^2k^2hw$ |
| Neural ePDOs (diag) | $c^2n/r + c^2(N+1)(N+2)n/2rp$ | $(c^2n^2/r + c^2n^2k^2/rp)hw + cnk^2hw$ |

## 7   EXPERIMENTS

### 7.1   MNIST-ROT

We first test our model on MNIST-rot dataset (Larochelle et al., 2007), which is a standard benchmark to test the equivariant models. The dataset contains $62k$ $28 \times 28$ randomly rotated gray-scale handwritten digits. Images in the dataset are split into 12k for training and 50k for testing.

As the images in the MNIST-rot dataset are orientation-unknown, we choose the group as $C_{16}$ for our model. Following the architecture in the Jenner & Weiler (2021) that consists of 6 steerable PDOs layers followed by two fully connected layers, we construct our model by replacing the last 5 steerable PDOs layers with our Neural ePDOs layers. More details about the model, training and hyperparameters analysis can be found in the supplementary material.

Our results can be found in Table 2. Some of our models use the regular field as intermediate feature fields and others use quotient representations (which are denoted by quotient in Table 2). Under the setting of using both finite difference and Gaussian derivatives, our model achieves significant

Table 2: Results in MNIST-rot. The test error with standard deviations are averaged over 5 runs.

| Method | Discretization | Test error (%) | Params |
|---|---|---|---|
| Vanilla CNN | – | 1.96±0.06 | 1.1M |
| Steerable PDOs | FD | 1.54±0.32 | 941K |
| Neural ePDOs | FD | **0.77±0.05** | **234K** |
| E2CNN(quotient) | – | 0.70±0.03 | 2.75M |
| Steerable PDOs (quotient) | Gauss | 0.74±0.04 | 951K |
| Neural ePDOs (quotient) | Gauss | **0.70±0.06** | **229K** |
| E2CNN | – | 0.72±0.03 | 2.69M |
| Steerable PDOs | Gauss | 0.75±0.02 | 941K |
| Neural ePDOs | Gauss | **0.65±0.04** | **234K** |
| E2CNN ($D_{16|5}C_{16}$) | – | 0.68±0.02 | 5.36M |
| Steerable PDOs ($D_{16|5}C_{16}$) | Gauss | 0.78±0.05 | 947K |
| Neural ePDOs ($D_{16|5}C_{16}$) | Gauss | **0.59±0.03** | **525K** |

Table 3: Results for ImageNet100. DA is short for data augmentation. The test error with standard deviations are averaged over 5 runs.

| Method | Discretization | DA | Test error (%) | Params |
|---|---|---|---|---|
| ResNet26 | – | × | 23.50± 0.46 | 13.12M |
| Res26 Steerable PDOs | FD | × | 19.92± 0.24 | 14.4M |
| Res26 Steerable PDOs | Gauss | × | 18.64± 0.29 | 14.4M |
| Res26 Neural ePDOs | FD | × | **17.76± 0.26** | **8.24M** |
| Res26 Neural ePDOs | Gauss | × | **17.44± 0.25** | **8.24M** |
| ResNet26 | – | ✓ | 15.11± 0.25 | 13.12M |
| Res26 Steerable PDOs | FD | ✓ | 13.94± 0.15 | 14.4M |
| Res26 Steerable PDOs | Gauss | ✓ | 12.54± 0.18 | 14.4M |
| Res26 Neural ePDOs | FD | ✓ | **12.08± 0.13** | **8.24M** |
| Res26 Neural ePDOs | Gauss | ✓ | **11.84± 0.14** | **8.24M** |

improvement over the steerable PDOs based model with only 1/4 parameters. It is noteworthy that the Gaussian derivatives method tends to achieve a better performance than the finite difference method for both steerable PDOs and our model, which is consistent to experiment results in Jenner & Weiler (2021). However, the performance of our quotient field model is inferior to the regular field, which is different from the results of steerable PDOs in Jenner & Weiler (2021). As is shown in Weiler & Cesa (2019), quotient fields can help to reduce redundancy in the regular fields in some cases. In our models, the redundancy is already less than steerable PDOs, hence, applying quotient fields in our model may hurt performance. To further demonstrate the potential capacity of our model, we enlarge the model size and employ $D_{16}$ regular field in the first five layers and restrict it to $C_{16}$ regular field in the last layer (results are denoted as $D_{16|5}C_{16}$ in Table 2). Our model improves the current SOTA model (Weiler & Cesa, 2019) while consuming only 9.8% of parameters.

## 7.2 EVALUATION ON NATURAL IMAGES

In general, the objects in real-world images are not always in a uniform orientation. So we believe that the models with rotation symmetry can generalize better on real-world images. ImageNet (Deng et al., 2009) is a large-scale dataset that consists of 1000 classes with roughly 1000 images per class, which is a common benchmark for image recognition. It contains 1.2 million training images and $50k$ validation images. Following Hou et al. (2019), we consider two experimental settings which correspond to different data scales. In the first setting, we conduct the experiments on a subset of ImageNet which randomly select 100 classes, and denoted it as ImageNet100. In the other one, we evaluate our model on the whole 1000 classes.
We choose ResNet-26 as the baseline model and construct equivariant PDO-based networks that are equivariant to $C_8$ group by taking place the convolution layer with $C_8$ equivariant module. We adopt regular representation for all the equivariant modules throughout the model. To make classification results rotation-invariant to the inputs, we add an orientation pooling (Cohen & Welling, 2016a)

Table 4: Results for ImageNet1k. The test error with standard deviations are averaged over 5 runs.

| Method | Discretization | Test error (%) | Params |
|---|---|---|---|
| ResNet26 | – | 26.4± 0.52 | 13.7M |
| Res26 Steerable PDOs | FD | 25.39± 0.35 | 15.02M |
| Res26 Steerable PDOs | Gauss | 23.55± 0.31 | 15.02M |
| Res26 Neural ePDOs | FD | **24.02±0.27** | **8.82M** |
| Res26 Neural ePDOs | Gauss | **23.15± 0.33** | **8.82M** |

before global average pooling. Notice that the $1 \times 1$ convolutions in ResNet26 are simply replaced with equivariant linear projection layers rather than the PDOs layer. Other convolution layers are replaced by steerable PDOs or Neural ePDOs which we denote as Res26 Steerable PDOs or Res26 Neural ePDOs. The output dimensions of each layer for the two models are scaled with the same factor to make its learnable parameters in Res26 Steerable PDOs comparable with the baseline. More detailed training settings can be found in the supplementary material.

Results on ImageNet100 and ImageNet1k are shown in the Table 3 and Table 4 respectively. In all the settings, our models significantly improve Steerable PDOs based models with fewer parameters (8.2M vs 14.4M) and computational costs (flops=$24.1$G vs $56.6G$). For ImageNet100, training with/without data augmentation is adopted. It is observed that equivariance can help improve the performance of the model and data augmentation can further enhance the performance of the equivariant model, which may be attributed to the approximate equivariance of the equivariant model incurred by discretization. In addition, we observe that performance improvement of Neural ePDOs over Steerable PDOs under no data augmentation setting is more significant. We conjecture our Neural ePDOs can help our model easier adapt to unseen patterns and hence tend to be more data efficient. It is also noticeable that the Gaussian discretization method still outperforms the finite difference method on the natural images.

## 8 CONCLUSION AND FUTURE WORKS

In this work, we propose a new nonlinear PDOs scheme that is both spatially adaptive and translation equivariance. A new equivariant theory is developed for our nonlinear PDOs scheme which gives a general equivariant formulation of it under any given symmetry group. The theory systematically characterizes the space of coefficient generators of our equivariant nonlinear PDOs for any given equivariance. Based on this theory, we efficiently implement it by adopting two-layer EMLP as the coefficient generators and, hence, name our model Neural ePDOs. Extensive experiments demonstrate our Neural ePDOs can significantly improve performance on MNIST-rot and ImageNet datasets. Particularly, we achieve new SOTA performance on MNIST-rot dataset with only 9.8% parameters compared to the previous best model.

To reduce the parameters and computational costs of generating coefficients, we have proposed efficient coefficient generators which are designed for features of regular field and quotient field. Efficient coefficient generators for other representation fields, *e.g.,* irreducible representation field, could be further explored.

It is worth emphasizing that the nonlinearity introduced in our adaptive PDOs has greatly improved their performance compared to linear PDOs even with smaller model sizes. There is still a large space to explore how to introduce nonlinearity into PDOs more efficiently.

Although we mainly focus on the two-dimensional plane, our framework can be readily extended to other homogeneous spaces such as spheres and 3D spaces. As our Neural ePDOs can achieve significant improvement on the two-dimensional image tasks, it is worth believing applications of Neural ePDOs in these domains are promising.

## ACKNOWLEDGMENT

Z. Lin was supported by National Key RD Program of China (2022ZD0160302), the major key project of PCL, China (No. PCL2021A12), the NSF China (No. 62276004), Qualcomm, and Project 2020BD006 supported by PKU-Baidu Fund.

**Reproducibility Statement**

The complete proof of the theorems is provided in supplementary material, and all the experimental details are provided in Section 7 and supplementary material.

**Ethics Statement**

The research in this paper does NOT involve any human subject, and our dataset is not related to any issue of privacy and can be used publicly. All authors of this paper follow the ICLR Code of Ethics (https://iclr.cc/public/CodeOfEthics).

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
