# OpenReview forum: "Neural ePDOs: Spatially Adaptive Equivariant Partial Differential Operator Based  Networks"
_ICLR.cc/2023/Conference — ICLR 2023 notable top 25%_

### Official Review · Reviewer_rHY6 · 2022-10-23

**Confidence:** 3
**Correctness:** 3
**Technical Novelty And Significance:** 3
**Empirical Novelty And Significance:** 2
**Recommendation:** 6

**Clarity, Quality, Novelty And Reproducibility:**

The paper is very well-written, and the message is clear. The idea builds on the prior works on ePDOs, but making it spatially adaptive is novel.

**Strength And Weaknesses:**

**Strength**
1. The paper is well-written and organized. It is pleasant to read.
2. The motivation of the proposed work is very clear: linear PDOs with translation equivariance is restricted to have shared coefficients. To achieve spatial adaptivity and maintain equivariance, nonlinear models are necessary.
3. Even though I have not read the details of the proofs, the theoretical results look natural and convincing.
4. In terms of implimentation, the authors reduce the model size by learning a diagonal coefficient matrix using an equivariant multilayer perceptron, which is novel.

**Weakness**
My concern of this paper mainly stems from the experiments
1. Compared to the prior work steerable PDOs, the proposed neural ePDOs, because of the additional eMLP for coefficient learning, will surely have a larger model size even when learning only a diagonal coefficient matrix. Can the authors clarify why there are less params for neural ePDOs compared to steerable PDOs in the tables?
2. Since the dataset use MNIST-rot (without flipping), what is the benefit of $D_{16}$ as the symmetry group?
3. This is related to the second question: can the authors report the result of E2CNN($C_16$) instead in table 1?
4. Can authors report the comparison of the actual training/testing *time* instead of the computational flops in the experiments?

**Summary Of The Paper:**

This paper proposes a spatially adaptive equivariant partial differential operator (ePDO) based networks. Motivated by the prior work on ePDO by Jenner & Weiler (2021), where a linear PDO is restricted to have constant coefficient matrix shared spatially for translation equivariance, the authors propose a nonlinear PDO scheme that is both spatially adaptive and translation equivariant. The spatially-adaptive coefficients in this work are instead learned based on the feature vector using a generator. When the symmetry group is an affine group acting on $\mathbb{R^2}$, constraints on the coefficients have been derived to ensure equivariance. The coefficient generation (subject to the derived equivariant constraint) is implemented using an equivariant multilayer perceptron. Numerical experiments are conducted to demonstrate the improved performance on MNIST-rot and Imagenet classification.

**Summary Of The Review:**

I think the paper is well-motivated and it has high quality in terms of both novelty and clarity. However, the experimental sections of the paper can be improved. I am willing to improve my rating based on the authors' response.

**Post author feedback**
The authors answered my concerns in detail. I am willing to increase the rating.

---

> ### Author Response · Authors · 2022-11-17
> **Response to Reviewer rHY6**
>
> Thank you very much for recognizing the novelty and clarity of our paper. We hope our answers can address your concerns.
>
> > Compared to the prior work steerable PDOs, the proposed neural ePDOs, because of the additional eMLP for coefficient learning, will surely have a larger model size even when learning only a diagonal coefficient matrix. Can the authors clarify why there are less params for neural ePDOs compared to steerable PDOs in the tables?
>
> A1: We give computational complexity analysis in the following table comparing the Neural ePDOs (diag) with both Neural ePDOs (full) and steerable PDOs.
> | Method | Param |Flops|
> |--|--|--|
> | Steerable PDOs | $c^2(N  + 1)(N  + 2)n/2$ | $c^2n^2k^2hw$ |
> | Neural ePDOs (full) | $c^2n/r+c^3(N+1)(N+2)n/2rp$  |  $(c^2n^2/r+c^3n^3k^2/rp)hw+ c^2n^2k^2hw$  |
> | Neural ePDOs (diag) | $c^2n/r+c^2(N+1)(N+2)n/2rp$  |  $(c^2n^2/r+c^2n^2k^2/rp)hw+ cnk^2hw$ |
>
> More details on the use of notations and analysis can be found in Section 6 of the revision.
>
> We can see, the Neural ePDOs (full) indeed require more computations compared to steerable PDOs, while Neural ePDOs (diag) requires significantly fewer parameters and flops than steerable PDOs by setting a proper r and p (rp>4 is enough). We can see r and p play important roles in reducing the parameters and flops of  Neural ePDOs (diag), in other words, diagonal restriction and EMLP design (bottleneck structure  (r) and partition (p) operation) together help to make our Neural ePDOs (diag) more efficient than steerable PDOs.
>
> >Since the dataset use MNIST-rot (without flipping), what is the benefit of  D16  as the symmetry group?
>
> A2: There is some misunderstanding and the model denoted as D16 is not globally equivariant to D16 group.These models follow architectures in Weiler&Cesa, in which group restriction operation is introduced. As has been presented in the last para of Section 6.1 (Section 7.1 in revision), these models are not fully equivariant to D16 and their feature fields would be restricted to C16 before the sixth layer. A more detailed explanation for use of D16 equivariant layer and group restriction layer can be found in  Weiler&Cesa. Here, we give a brief description. Although the MNIST-rot dataset presents global rotational symmetry, it also shows local rotational and reflectional symmetries. For models only equivariant to global symmetry, they lack the ability to generalize over these local symmetries. However, if we adopt the full symmetry models,  their global invariance would lead to information loss. To deal with this problem, the group restriction operation is carried out to make models locally equivariant but globally invariant only to the symmetry presented in the data. To make the description clearer, we use $D_{16|5}C_{16}$ instead of $D_{16}$ to denote these models in the revision.
>
> >  Can the authors report the result of E2CNN(C16) instead in table 1?
>
> A3: We have added the E2CNN($C_{16}$) results in the revision. The results are cited from competitive MNIST rot experiments in Weiler&Cesa. Our model still outperforms the E2CNN in this case with significantly fewer parameters.
>
> > Can authors report the comparison of the actual training/testing time instead of the computational flops in the experiments?
>
> A4: We show the training and testing time in the following table. We train and test models on single GTX 1080 Ti. The training time is obtained by training model for 1 epoch, and the test time is the inference time for one image.
>
> |Model  | Testing time(s) |Training time(s)|
> |--|--|--|
> | Steerable PDOs | 0.0194 | 50.25 |
> | Neural PDOs | 0.0201  |  102.45 |
>
> As shown in table, the gap between the test time of Neural ePDOs and steerable PDOs is minor. However, the training time of steerable PDOs is shorter, despite Neural ePDOs having fewer FLOPs.  It is because the steerable PDOs is implemented using convolution operators, which are highly optimized in existing speedup libraries. In comparison, we just give a naive implementation of Neural ePDOs without any support from these libraries. Of course, a specific CUDA kernel can be implemented for GPU acceleration to further speed up the training process for Neural ePDOs, but this is beyond the research and we leave this as future work. We have added this part in Section K in the supplementary materials.
>
>
> Weiler, M., & Cesa, G. (2019). General e (2)-equivariant steerable CNNs. Advances in Neural Information Processing Systems, 32.

---

> > ### Comment · Reviewer_rHY6 · 2022-12-06
> > **Thank you for the response**
> >
> > Thank you for the detailed response. I am willing to increase my rating to weak accept.

---

### Official Review · Reviewer_vWQG · 2022-10-24

**Confidence:** 4
**Clarity, Quality, Novelty And Reproducibility:** See above.
**Correctness:** 4
**Technical Novelty And Significance:** 3
**Empirical Novelty And Significance:** 3
**Recommendation:** 8

**Strength And Weaknesses:**

I am impressed by the designed method. The authors design a novel equivariant MLP as the neural ePDO to approximate the PDO with translation equivariance. This idea is novel and makes sense.

The authors theoretically verify the translation equivariance of the proposed neural ePDOs.

The authors also conducted comprehensive experiments on MNIST and ImageNet. The results show the proposed method significantly outperforms existing methods.

My major concern is on the literature review and comparison. The authors are encouraged to well position the present work in the existing works.

**Summary Of The Paper:**

This paper designs a novel neural ePDO. PDO has wide applications but needs linearity to ensure the translation equivariance. This paper proposes to use an equivariant MLP as the neural ePDO to realize the translation equivariance. The authors give comprehensive theory and experiments.

**Summary Of The Review:**

Overall, I recommend accept.

---

> ### Author Response · Authors · 2022-11-17
> **Response to Reviewer vWQG**
>
> Thank you very much for recognizing the results and novelty of our work. Here are the replies to your concerns.
>
> > My major concern is on the literature review and comparison. The authors are encouraged to well position the present work in the existing works.
>
> A1: Thanks for your suggestion. Our work follows the approach of steerable CNNs, and we have clarified this missing point in the revision. Besides, we have supplied a detailed comparison between our method and steerable PDOs in Section I  in supplementary materials in the revision.

---

### Official Review · Reviewer_j4DB · 2022-10-26

**Confidence:** 4
**Correctness:** 3
**Technical Novelty And Significance:** 3
**Empirical Novelty And Significance:** 3
**Recommendation:** 8

**Clarity, Quality, Novelty And Reproducibility:**

The paper could use some editing and proof-reading.  There are some typos and grammatical mistakes.  The introduction is also fairly long and vague.  More seriously, there are some fairly broad and undefended claims in the exposition which should be removed, softened or cited.
- "our model is much less redundant than steerable PDOs"
- "As in practice, regular representation and quotient representation (see supplementary material) are
mostly adopted for equivariant networks for their superior performance."
- "which can be regarded as a set of functional equations that are not easy to solve."
- "the mappings between smooth functions are usually defined as partial differential
operators (PDOs) such as Laplacian or divergence in the physics area"
- "From the perspective of gradients, the parameters of the PDOs are updated by the globally pooled loss gradients, which results in sub-optimal feature learning at each position."

There does not appear to be code, but the complete proofs, method details, and experimental details are provided and so the results could reasonably be reproduced.

Though relatively similar to PDO-eConv and Steerable PDO, this method adds non-constant coefficients to the PDOs which requires generalizing the equivariance constraint.  It is novel.

**Strength And Weaknesses:**

### Strengths
- PDO based networks have been considered by several authors recently including equivariant versions and its reasonable and novel to consider an extension with non-constant coefficients.  In order to maintain shift equivariance, the coefficients must be functions of input function.  So $ f (\partial_x f)$ is okay but $x (\partial_x f)$ is not.
- The authors do a good job deriving the equivariance constraints for the coefficient networks in both the full and diagonal case.
- The experimental results are pretty convincing. The most important baseline is Steerable PDO and that is compared to.  Variations such as discretization, different intermediate representation types, and data augmentation are considered.  In both experiments the current method outperforms using fewer parameters than baselines.

### Weaknesses / Questions
- Some things are missing from the experiments.  Variance should be reported over multiple runs.  How are the hyperparameters tuned?  It is not clear they were.  How does performance of the model and baseline vary as the number of parameters vary?    I'd also like to see a non-equivariant ablation using non-linear PDOs but without the equivariance constraint.
- The diagonal operators are claimed to be more efficient.  I'd like to see how much more efficient.  That is, I'd like to see a comparison to the largest practical model which can be built using full matrices.
- The experiments are both image classification.  Given that the motivation for PDOs is in differential equations, it would nice to see an application for modeling differential equations or a non-invariant application.  One is left wondering what specifically about PDOs is important for solving vision tasks?  If I replaced the FD $\partial^{\mathbf{i}}$ operator with other fixed operators with similar statistics would I get similar performance?  If so, our understanding for why this network is effective changes.  Perhaps, it is merely that the operators $\partial^{\mathbf{i}}$ provide a more efficient basis for the space of spatial kernels than elementary matrices and nothing to do with their meaning as PDOs.
- In 5.1, it says $\rho = p \rho_0$.  If true, this seems very constrained.  It would be good to do a comparison to other choices.  That said, I don't really understand where $p \rho_0$ is actually used in the architecture since the experiments section refers to using regular and quotient representations.
- Is it clear the diagonal assumption is not overly restrictive?  The experiments offer good evidence it is not, but it would be nice to see theoretical evidence as well.

### Minor Points
6.1 Para 2, Incorrect Reference to Weiler & Cesa for PDOs.
Eqns. 9, 10.  The notations for how the tensor product representations act on the coefficients are a bit unclear.
Page 4 "discription" -> description
Page 3 "semi-product" -> "semi-direct product"

**Summary Of The Paper:**

This paper introduces an equivariant neural network architecture based on partial differential operators (PDOs) in the layers.  These operators are constrained to be translation and rotation equivariant.  Relative to previous work, PDO-eConv (Shen et al, 2020) and Steerable PDO (Jenner & Weiler , 2021), this work adds non-constant coefficients to the PDOs.  This changes the equivariance constraint relative to prior work and the authors work out the new equivariance constraint.  They also provide a version of the constraint under the assumption the operators are diagonal.  Experiments show the architecture is more efficient than Steerable PDO as it attain better accuracy on rotated MNIST and ImageNet using fewer parameters.

**Summary Of The Review:**

This paper makes a novel and reasonable generalization of previous equivariant PDO work.  The derivation and method seem sound. The experimental results seem quite strong although not as thorough as they could be.  The choice of experimental domains is limited to invariant classification task.  The paper presentation and polish could use work.

---

> ### Author Response · Authors · 2022-11-17
> **Response to Reviewer j4DB (part1)**
>
> Thank you very much for viewing our work novel and sound. We hope our answers can address your concerns.
>
> >Variance should be reported over multiple runs.
>
> A1: Thanks for pointing out this missing point. All the results we reported in the paper are the average of 5 runs with random seeds, and we have added the variance in the revision.
>
> >How are the hyperparameters tuned? How does performance of the model and baseline vary as the number of parameters vary?
>
> A2: Here, we show the hyperparameter analysis of our model in the following table. The number of our models' parameters is mainly controlled by the partition number $p$ and the reduction ratio $r$, so we vary these two parameters respectively.
>
> |  | $p=\frac{z}{8}$ |$p=\frac{z}{4}$|$p=\frac{z}{2}$ |
> |--|--|--|--|
> | r=1 | 0.62$\pm$ 0.05 |  0.65$\pm$ 0.04 |  0.67$\pm$ 0.03  |
> | r=2 | 0.66$\pm$ 0.06  |  0.64$\pm$ 0.03 |   0.69$\pm$ 0.03 |
> | r=4 |  0.66$\pm$ 0.04 |  0.67$\pm$ 0.03 |  0.71$\pm$ 0.04  |
>
> All the results with standard deviations are averaged over 5 random runs with random seeds. $z$ is the number of input fields of the current layer. Besides, we also show a scatter plot in Figure 1 in the supplementary. From the table and figure,  we can see that all the Neural ePDOs models outperform steerable PDOs with significantly fewer parameters. Besides, as the size of Neural ePDOs is large enough, the performance gain is very limited, which demonstrates such a design can help to trade off efficiency and accuracy. We have added the hyperparameter analysis in Section J in supplementary materials.
>
> >I'd also like to see a non-equivariant ablation using non-linear PDOs but without the equivariance constraint.
>
> A3: Here, we give the results of the non-equivariant Neural PDOs on MNIST-rot experiment. We expand the channels to 64—>120—>120—>160—>160—> 240 while fixing other hyperparameters. The results are shown as follows:
>
> |  Models | Test error |Params|
> |--|--|--|
> | Vanilla CNN | 1.96$\pm$0.06 | 1.1M |
> | Neural PDOs | 1.49$\pm$ 0.07  |  0.26M  |
> | Neural ePDOs($C_{16}$) | 0.65$\pm$0.04  |  0.23M  |
>
> We can see from the table that the non-equivariant Neural PDOs can achieve better performance than Vanilla CNN with fewer parameters. In addition, incorporating symmetries can further improve performance.
>
> >The diagonal operators are claimed to be more efficient. I'd like to see how much more efficient. That is, I'd like to see a comparison to the largest practical model which can be built using full matrices.
>
> A4:  We give computational complexity analysis in the following table comparing the Neural ePDOs (diag) with both Neural ePDOs (full) and steerable PDOs.
> | Method | Param |Flops|
> |--|--|--|
> | Steerable PDOs | $c^2(N  + 1)(N  + 2)n/2$ | $c^2n^2k^2hw$ |
> | Neural ePDOs (full) | $c^2n/r+c^3(N+1)(N+2)n/2rp$  |  $(c^2n^2/r+c^3n^3k^2/rp)hw+ c^2n^2k^2hw$  |
> | Neural ePDOs (diag) | $c^2n/r+c^2(N+1)(N+2)n/2rp$  |  $(c^2n^2/r+c^2n^2k^2/rp)hw+ cnk^2hw$ |
>
> More details on the use of notations and analysis about the above table can be found in the Section 6 of revision.
>
> We can see, the Neural ePDOs (full) indeed require more computations compared to steerable PDOs, while Neural ePDOs (diag) requires significantly fewer parameters and flops than steerable PDOs by setting a proper r and p (rp>4 is enough). We can see r and p play important roles in reducing the parameters and flops of  Neural ePDOs (diag), in another word, diagonal restriction and EMLP design (bottleneck structure  (r) and partition (p) operation) help to make our Neural ePDOs (diag) more efficient than steerable PDOs.

---

> > ### Author Response · Authors · 2022-11-17
> > **Response to Reviewer j4DB (part 2)**
> >
> > >it would nice to see an application for modeling differential equations or a non-invariant application. One is left wondering what specifically about PDOs is important for solving vision tasks? If I replaced the FD ∂i operator with other fixed operators with similar statistics would I get similar performance? Perhaps, it is merely that the operators ∂i provide a more efficient basis for the space of spatial kernels than elementary matrices and nothing to do with their meaning as PDOs.
> >
> > A5: We have carried out the DeepCFD (modeling Navier-Stokes equation) experiment following the settings in Jenner & Weiler in which the output is composed of a scalar field and vector field (not invariant). We replace the steerable PDOs with our Neural ePDOs. The results are as follows:
> > |Model  | MSE |
> > |--|--|
> > | Steerable PDOs | 2.32 $\pm$ 0.08 |
> > | Neural ePDOs | **2.11 $\pm$ 0.07**  |
> >
> > Our model still outperforms steerable PDOs in this experiment.
> >
> > Large amounts of PDOs have been widely proposed and applied in the physics area, such as Navier-Stokes equation for the description of flow and the Schrödinger equation for the description of microscopic particles,  which can give researchers inspiration to design new operators in vision tasks. In addition, a lot of prior works have shown PDOs achieved great performance on various vision tasks (Jiang 2019, Shen 2021, Finzi 2021, Wiersma 2022). Especially, PDOs, for their locality, are very suitable for processing unstructured data (Jiang 2019, Shen 2021) which need local parameterization.  We agree that it is interesting to see the performance of the network when replacing the FD ∂i operator. However, the main objective of the paper is developing equivariant nonlinear PDOs to enhance the capacity of steerable PDOs, the replacement of other operators would not affect our main conclusion. Of course, the method developed in this paper to endow PDOs with spatial adaptivity and equivariance can also provide some insights for designing other nonlinear operators. This could be further exploited in the future, and it is out of the scope of the paper.
> >
> > > That said, I don't really understand where pρ0 is actually used in the architecture since the experiments section refers to using regular and quotient representations.
> >
> > A6:  Actually, the $\rho$ here is not refer to the single regular or quotient representation but the representation of the whole input field. It is stacked by $n$ regular or quotient representations. In our paper, we divide the whole representation into $p$ partitions uniformly such that each partitions $\rho_0$ are the stack of $\frac{n}{p}$ regular or quotient representations. Then, we restrict the output of each partitions to be the same, which is one of the key points for enhancing efficiency. As you can see in Section J in supplementary materials, increasing $p$ would not lead to a significant performance drop, which can help to trade off accuracy and efficiency.
> >
> > > Is it clear the diagonal assumption is not overly restrictive? The experiments offer good evidence it is not, but it would be nice to see theoretical evidence as well.
> >
> > A7: In fact, our diagonal assumption is not overly restrictive. We provide a theoretical insight from the perspective of information interaction. It seems that the diagonal coefficient matrices can not directly aggregate the information along channel dimension, however, in the coefficient generation process, the information between channels has already interacted. To be specific, the information interaction process is divided into two steps. First, the generation process of the PDOs aggregates information along channels as each independent fiber is the input of the coefficient generator. Then, the PDOs' action on feature fields implicitly disperses the encoded channel-wise information to the spatial domain along with the aggregation of spatial information. Compared with Steerable PDOs and CNNs which mix the two steps,  our method can thus reduce the heavy redundancy in the network.
> >
> > Besides, the minor points and writing issues have been fixed in the revision.
> >
> >
> > Jenner, E., & Weiler, M. (2021). Steerable Partial Differential Operators for Equivariant Neural Networks. In International Conference on Learning Representations
> >
> > Marc Finzi et al. (2021). Probabilistic Numeric Convolutional Neural Networks. International Conference on Learning Representations.
> >
> > Shen Zhengyang et al. (2021) "PDO-eS2CNNs: Partial Differential Operator Based Equivariant Spherical CNNs." Proceedings of the AAAI Conference on Artificial Intelligence.
> >
> > Jiang, C. M et al. (2019). Spherical CNNs on Unstructured Grids. In International Conference on Learning Representations.
> >
> > Wiersma, R. et al. (2022). DeltaConv: anisotropic operators for geometric deep learning on point clouds. ACM Transactions on Graphics (TOG)

---

> > > ### Comment · Reviewer_j4DB · 2022-12-01
> > > **Thank you for the response**
> > >
> > > This is quite a detailed and thorough response to my concerns.  I  believe it warrants and increase in score.
> > >
> > > For question of diagonal efficiency, I think it is helpful to have the complexity analysis.  I'd also like to see a performance comparison as a way of demonstrating the efficiency.  I.E. for the same number of parameters or flops how much worse is the performance of Neural ePDO(full) vs. Neural ePDO(diagonal).

---

> > > > ### Author Response · Authors · 2022-12-11
> > > > **Additional response to Reviewer j4DB**
> > > >
> > > > We sincerely thank you for the recognition of our response. For the performance comparison between Neural ePDOs (full) and Neural ePDOs (diagonal), we construct Neural ePDOs (full) model by replacing  Neural ePDOs (diagonal) layers with Neural ePDOs (full) layers. For a fair comparison, we keep model parameters almost the same and other training settings unchanged. The following table shows the results.
> > > >
> > > > |  | Test error |Params|
> > > > |--|--|--|
> > > > | Neural ePDOs (diagonal)| **0.65$\pm$ 0.04** |	234K |
> > > > | Neural ePDOs (full) | 0.75$\pm$ 0.05  |	  248K	|
> > > >
> > > >
> > > > We can see that the diagonal model achieves better performance, which further confirms the efficiency of such design.

---

### Official Review · Reviewer_V42w · 2022-10-26

**Confidence:** 4
**Correctness:** 4
**Technical Novelty And Significance:** 3
**Empirical Novelty And Significance:** 3
**Recommendation:** 8

**Clarity, Quality, Novelty And Reproducibility:**

As mentioned above the paper could benefit from additional details and intuitive explanations, though overall the paper is sound.

The work is of high quality and relies on state-of-the-art techniques, the experiments are appropriate.

The work is novel.

In regards to reproducibility, all theoretical details as well as an appendix with details are provided, but releasing code with this submission would greatly improve reproducibility (translating the work to code may be a challenge!)

**Details Of Ethics Concerns:**

No concerns

**Strength And Weaknesses:**

**Strengths**

1. The paper implements an intuitive and appealing idea: making the main layers of NN non-linear and adaptive to the present features locally. In some sense this also happens in transformers via attention, or via deformable convolutions [Dai] and possibly other works, however, what makes this approach interesting is that is efficient in that the operations (PDOs) are strictly local. Their non-linear adaption boilos down to a point-wise (equivariant) MLP.
2. The proposed work is timely, and builds up recent advances in PDO-based deep learning.
3. The experiments are well done and clearly underpin the benefit of non-linear ePDOs vs linear ePDOs and normal resnets.
4. The paper is theoreticall sound.
5. The paper is overall well written and precise, but could occasionally benefit form added details and intuition.

Dai, J., Qi, H., Xiong, Y., Li, Y., Zhang, G., Hu, H., & Wei, Y. (2017). Deformable convolutional networks. In Proceedings of the IEEE international conference on computer vision (pp. 764-773).

**Weaknesses**

1. Upon first reading, it is hard to distill the differences between the ePDO of Jenner and Weiler, from the contributions in this paper. It is clear though that it is the modification of the PDO coefficient being dependent on the input (which is an important contribution in itself!), however, the kernel constraint is very hard to digest. It would have been nice to put this in perspective relative to the original constraint posed in the paper by Jenner and Weiler.
2. The paper is a bit untransparent when it comes to the "steerabble" aspect of things. Parts are clear, vector that collects derivatives can in principle be steered by some representation of G, like in Eq.5, but it is unclear what the types of the input and output feature fields are, and how to set them. It may help a lot of some intuition is provided on how to pick those reps, also in relation to the kernel constraint. Overall, I think an intuitive breakdown of Eq 8 would be helpful.
3. Some minor details are missing:
    1. to which sub-group is the quotient representation defined?
    2. The method is overall more efficient (in memory and flops), could it be made more precise where this effiency comes from? The operations themselves are more expensive than the original ePDOs, right? Is it mainly due to working with less independent feature channels (e.g. choosing a lower p in the non-linear case compared to the linear ePDO setting)?
    3. In proposition 3, why is it important to consider regular or quotient representations (which are also regular), and not irreducible representations? I.e., why not formulate it for any representation of G?
    4. Might this have to do with sec 5.1 where the equivariant MLPs use ReLUs. If working with irreps, not all activation functions may be allowed (though this seems to have nothing to do with proposition 3 otherwise).
    5. What sigma did you pick for the Gaussian derivatives?

**Summary Of The Paper:**

The paper describes an equivariant implementation of non-linear partial differential operator (PDO) based networks. A key contribution of the paper is in the non-linear adaption of equivariant PDOs (ePDOs), which makes them more parameter efficient and expressive. The idea is intuitive and sensible, based on the idea that the operators should be locally adapted to local patterns. This is achieved by letting a local feature vector determine the PDO coefficients locally. This coefficient predictor then has to be equivariant as well, and this is achieved by equivariant MLPs. The paper then provides a clear recipe/theory for the conditions under which the PDO is indeed equivariant, which guides the construction of an efficient implementation. The proposed method is extensively validated on rotated MNIST as well as on ImageNet, and confirm the added benefit of working with non-linear ePDOs as opposed to linear ePDOs.

**Summary Of The Review:**

The paper is a great submission to ICLR; it is novel and sound, but could still benefit from additional details and intuitive explanations. I judge these improvement to be possible within the rebuttal period/before the cam-ready as they can mostly be fixed with textual improvements. I therefore recommend accept.

---

> ### Author Response · Authors · 2022-11-17
> **Response to Reviewer V42w (part1)**
>
> Thank you very much for appreciating our work and giving us sincere suggestions. The following replies endeavor to address your concerns.
>
> >It would have been nice to put this in perspective relative to the original constraint posed in the paper by Jenner and Weiler.
>
> A1: In Jenner & Weiler, they first make use of the duality between polynomials and PDOs to construct an isomorphic map from PDOs to a matrix of polynomials, which transforms the problem of finding equivariant PDOs into the problem of solving polynomials based equivariant kernel. Then, they solve the polynomials based equivariant kernel by extending the solution basis of the equivariant kernel in Weiler&Cesa to the polynomials form. However, such a method can not be easily extended to the non-linear PDOs proposed in our work, because there is no such isomorphic map to transform non-linear PDOs to the matrix of polynomials. In comparison, in our paper, we directly deduce the equivariant constraint on the coefficients of PDOs. Note that the solution space of steerable PDOs is exactly included in the solution space of Eq.(9), as we choose the coefficient generator to be the constant function of the input. In addition, compared to the method in Jenner & Weiler, we no longer need to compute the solution basis for each irreducible representation pair manually beforehand, because the solving process is totally automatic for any representation pairs under our framework, which makes it much easier to extend to $G \leq O(n)$ for any $n\in \mathbb{N}$. We have added a comparison part of our work and Jenner & Weiler in the supplementary materials in the rebuttal revision.
>
> >It may help a lot of some intuition is provided on how to pick those reps, also in relation to the kernel constraint. Overall, I think an intuitive breakdown of Eq 8 would be helpful.
>
> A2: Proposition 1 (Eq8) and proposition 2 (Eq9) are the general theory of equivariant form of our non-linear PDO, and any representation type can be chosen for the input and output fields.  We have emphasized this in Section 4.2 in the rebuttal revision. The choices of the representation type of equivariant network have been well studied in Weiler & Cesa. Especially, they find that regular and quotient representations tend to achieve better performance.
>
> In fact, the Eq(8) is the direct constraint on the coefficient generators of their original form, which is a little bit abstract. So in order to uncover the intrinsic structure of the coefficient generators, we propose proposition 2 which states that the coefficient generators, under slight modification, are essentially equivariant maps.
>
> >To which sub-group is the quotient representation defined?
>
> A3: We have shown the form of the quotient representation we chose in Section G para 2 in supplementary.
>
> >The method is overall more efficient (in memory and flops), could it be made more precise where this effiency comes from?
>
> A4:  We give a computational complexity analysis in the following table comparing the Neural ePDOs (diag) with both Neural ePDOs (full) and steerable PDOs.
> | Method | Param |Flops|
> |--|--|--|
> | Steerable PDOs | $c^2(N  + 1)(N  + 2)n/2$ | $c^2n^2k^2hw$ |
> | Neural ePDOs (full) | $c^2n/r+c^3(N+1)(N+2)n/2rp$  |  $(c^2n^2/r+c^3n^3k^2/rp)hw+ c^2n^2k^2hw$  |
> | Neural ePDOs (diag) | $c^2n/r+c^2(N+1)(N+2)n/2rp$  |  $(c^2n^2/r+c^2n^2k^2/rp)hw+ cnk^2hw$ |
>
> More details on the use of notations and analysis can be found in Section 6 of the revision.
>
> We can see, the Neural ePDOs (full) indeed require more computations compared to steerable PDOs, while Neural ePDOs (diag) require significantly fewer parameters and flops than steerable PDOs by setting a proper r and p (rp>4 is enough). We can see r and p play important roles in reducing the parameters and flops of  Neural ePDOs (diag), in other words, diagonal restriction and EMLP design (bottleneck structure (r) and partition (p) operation) help to make our Neural ePDOs (diag) more efficient than steerable PDOs.

---

> > ### Author Response · Authors · 2022-11-17
> > **Response to Reviewer V42w (part 2)**
> >
> > >In proposition 3, why is it important to consider regular or quotient representations (which are also regular), and not irreducible representations?
> > >If working with irreps, not all activation functions may be allowed
> >
> > A5: As pointed out by Weiler&Cesa, the regular or quotient representation models significantly outperform the irrep models on various datasets, which shows their great expressive power in practice. So in our paper, we only consider efficient structure (proposition 3) for such representations and choose them for experiments as Jenner & Weiler. Of course, our general theory (proposition 2) can handle any representation type of G, and the irrep can also be efficiently implemented, e.g, restricting the coefficient matrices to block diagonal matrices. We leave searching for the most efficient form for irreducible representation as future work.
> >
> > > What sigma did you pick for the Gaussian derivatives?
> >
> > A6: Thanks for figuring out this missing point, we choose sigma as 1 in our experiment. We have added this setting at the reversion of supplementary materials.
> >
> > Weiler, M., & Cesa, G. (2019). General e (2)-equivariant steerable cnns. _Advances in Neural Information Processing Systems_, _32_.
> >
> > Jenner, E., & Weiler, M. (2021). Steerable Partial Differential Operators for Equivariant Neural Networks. In _International Conference on Learning Representations_

---

> > > ### Comment · Reviewer_V42w · 2022-11-21
> > > **Great response, great work**
> > >
> > > Thank you for your detailed response. The clarifications (here on openreview and in the revised version) are much appreciated! I stand by my original score of an 8, the response further comfirms my believe in this positive rating.

---

### Decision · Program_Chairs · 2023-01-20

**Decision:**

Accept: notable-top-25%

**Justification For Why Not Higher Score:**

Although the theoretical extensions are significant, the empirical results show marginal improvements compared to prior works.

**Justification For Why Not Lower Score:**

The paper presents novel extensions to neural PDO that are recognized by all the reviewers.

**Metareview: Summary, Strengths And Weaknesses:**

This paper introduces equivariant neural network-based partial differential operators (PDOs). The prior works in this space assume that weight matrixes in each layer are shared across different locations to guarantee translation equivariance.  However, this submission lifts this assumption by introducing spatially adaptive neural PODs and it shows that additional equivariance (e.g., rotation equivariance) can be obtained as long as the coefficient-generating MLPs are itself equivariant. The proposed technique is analyzed theoretically and the empirical results support the efficacy of the introduced layers. All the reviewers acknowledged the novelty of this submission, the extensivity of empirical and theoretical studies, and the clarity of writing. There were minor issues in the presentation that were resolved during the rebuttal. Given this, I am happy to recommend accept.

**Note From Pc:**

if the above contains the word "oral" or "spotlight" please see: "oral" presentation means -> notable-top-5% and "spotlight" means -> notable-top-25%. As stated in our emails, we are disassociating presentation type from AC recommendations

**Summary Of Ac-Reviewer Meeting:**

N/A